# Nanoemulsions as a Promising Carrier for Topical Delivery of Etodolac: Formulation Development and Characterization

**DOI:** 10.3390/pharmaceutics15102510

**Published:** 2023-10-23

**Authors:** Samet Özdemir, Burcu Üner, Alptuğ Karaküçük, Burak Çelik, Engin Sümer, Çetin Taş

**Affiliations:** 1Department of Pharmaceutical Technology, Faculty of Pharmacy, Istanbul Health and Technology University, 34445 Istanbul, Turkey; 2Department of Administrative and Pharmaceutical Sciences, University of Health Science and Pharmacy in St. Louis, St. Louis, MO 63110, USA; burcu.uner@uhsp.edu; 3Department of Pharmaceutical Technology, Faculty of Pharmacy, Ankara Medipol University, 06050 Ankara, Turkey; alptug.karakucuk@ankaramedipol.edu.tr; 4Department of Pharmaceutical Technology, Faculty of Pharmacy, Bezmialem Vakif University, 34093 Istanbul, Turkey; bcelik@bezmialem.edu.tr; 5Experimental Research Center (YUDETAM), Faculty of Medicine, Yeditepe University, 34755 Istanbul, Turkey; engin.sumer@yeditepe.edu.tr; 6Department of Pharmaceutical Technology, Faculty of Pharmacy, Yeditepe University, 34755 Istanbul, Turkey; cetin.tas@yeditepe.edu.tr

**Keywords:** etodolac, nanoemulsions, topical drug delivery, paw edema, permeation

## Abstract

This research primarily focuses on the development of innovative topical nanoemulsions for etodolac, aimed at surmounting its inherent limitations. The preparation of etodolac nanoemulsions is accomplished through a combination of high shear homogenization and ultrasonication methods. The optimization of the formulation components is systematically conducted using the design of experiments methodology. The droplet size (DS), polydispersity index (PDI), and zeta potential (ZP) of the optimized formulation were assessed using the differential light scattering (DLS) technique. Surface morphology examinations were conducted using electron microscopy, while interactions between excipients and the drug were analyzed through FTIR analysis. Additionally, in vitro release and ex vivo permeability studies were carried out. Furthermore, anti-inflammatory activity was evaluated in the context of a carrageenan-induced paw edema model in rats. The DS, PDI, and ZP of the optimal formulation were 163.5 nm, 0.141, and −33.1 mV, respectively. The in vitro release profile was assessed as a sustained release by following a non-Fickian drug transport. The flux of etodolac nanoemulsions and coarse dispersions were 165.7 ± 11.7 µg/cm^2^ h and 59.7 ± 15.2 µg/cm^2^ h, respectively. Enhanced edema inhibition was observed at 13.4%, 36.5%, and 50.65% for the 6th, 8th, and 24th hours, respectively. Taken together, these results confirmed that nanoemulsions are promising carriers for the topical delivery of etodolac.

## 1. Introduction

Topical drug delivery has emerged as a viable alternative to conventional oral and parenteral routes, offering numerous advantages such as a localized treatment, reduced systemic side effects, a prolonged therapeutic effect, and improved patient compliance [1]. Despite these benefits, the *Stratum corneum*, the outermost layer of the skin, presents a formidable barrier for the drug permeation [2].The physicochemical properties (e.g., high melting points, hydrophilicity, or large molecular sizes) of drugs can limit their suitability for topical delivery. The main objective of extensive research is to overcome these limitations and improve the delivery of drugs through the skin. This drives the investigation of new formulations and delivery methods.

In this context, nanoemulsions have attracted significant attention in topical drug delivery. They provide a versatile and efficient platform for improving skin penetration and precisely releasing therapeutic agents. These innovative delivery systems, made of tiny oil or water droplets stabilized by surfactants or co-surfactants, showcase outstanding physicochemical properties suitable for topical use [3]. The small droplet size of nanoemulsions provides a large interfacial area, facilitating improved solubilization and dissolution rates of hydrophobic drugs, such as those with a limited aqueous solubility [3,4]. This characteristic enables the efficient incorporation and permeation of lipophilic drugs through the skin’s lipid barrier, enhancing their bioavailability and therapeutic efficacy. Hence, utilizing nanoemulsions for topical drug delivery holds great promise in overcoming the challenges associated with conventional drug delivery methods.

An example of a drug facing topical delivery challenges is etodolac, a nonsteroidal anti-inflammatory drug (NSAID) known for its strong analgesic and anti-inflammatory effects in the treatment of rheumatoid arthritis [5]. It is classified as a biopharmaceutical classification system (BCS) class II drug that is characterized by a poor aqueous solubility and high permeability [6,7]; however, its poor aqueous solubility and the formidable barrier posed by the skin hinder its effective topical permeation. Due to the relatively short duration of action in the body, oral administration of the medication requires frequent dosing; however, this increased dosing frequency can lead to adverse cardiovascular effects, fluid retention, edema, and gastrointestinal problems, including bleeding, ulceration, and perforation [7,8,9]. Recently, there has been a growing interest to overcome these challenges, and local or topical formulation strategies utilizing nano-sized systems have gained significant attention [6,7,8,9,10]. By encapsulating etodolac within a nanoemulsion system, it is possible to enhance its solubility, increase its penetration through the skin, and ultimately improve its therapeutic efficacy. This innovative approach showcases the potential of nanoemulsion-based drug delivery systems to address the limitations of conventional drug delivery, ultimately improving patient outcomes.

Numerous parameters play a significant role in the design of nanoemulsions considered as an alternative to the conventional dosage form of etodolac. Designing nanoemulsions requires the precise selection of ingredients to achieve effective drug incorporation, targeting, and desired release patterns. It is crucial to optimize factors like the ratio of oil to water, the ratio of surfactant to co-surfactant, and the techniques used for mixing to attain a stable and efficient nanoemulsion formulation [11]. To streamline this process, the design of experiment (DOE) approach offers a systematic and efficient methodology. By employing DOE, multiple factors and their interactions can be assessed simultaneously, leading to a better understanding of the formulation’s critical quality attributes (CQAs) [12]. In this approach, various factors such as surfactant type and concentration, oil phase composition, and processing conditions are carefully selected and varied according to a predefined experimental design matrix. By systematically studying the effects of these factors on the response variables of interest, such as droplet size, polydispersity, and physical stability, an optimized formulation can be identified. Statistical analysis of the experimental results helps identify significant factors and their interactions, enabling the formulation scientist to make data-driven decisions and fine-tune the nanoemulsion formulation for the desired properties. The DOE approach significantly reduces the number of experiments required and provides valuable insights for formulation optimization, leading to the development of stable and efficient nanoemulsions of etodolac [11].

The present investigation aims to formulate an optimal nanoemulsion of etodolac to enhance its skin penetration. Key characteristics of the formulation and process parameters were monitored, then the DOE approach was employed to optimize the nanoemulsion formulation. The interaction between the formulation variables was investigated and the droplet size, droplet size distribution, zeta potential, and encapsulation efficiencies were evaluated as the response variables. After physicochemical characterization studies of the optimal nanoemulsion formulation in vitro, ex vivo and in vivo studies were performed.

## 2. Materials and Methods

### 2.1. Materials

The etodolac was kindly provided by Nobel Pharmaceuticals (Istanbul, Turkey). Caprylic/capric triglyceride (Crodamol™ GTCC), isopropyl isostearate (Crodamol™ IPIS), isopropyl myristate (Crodamol™ IPM) and isopropyl palmitate (Crodamol™ IPP) were obtained from Croda Turkey (Istanbul, Turkey). Castor oil, olive oil, sesame oil, Poloxamer^®^ 188 and Poloxamer^®^ 407, Brij^®^ 35 and tyloxapol were supplied by Sigma-Aldrich (Schnelldorf, Germany). Tego Care^®^ 450 was provided by Evonik Industries (Essen, Germany). All the chemicals and reagents used in the formulation, production, and analysis stages were of analytical grade.

### 2.2. Methods

#### 2.2.1. Selection of Excipients

An oil screening study was performed based on a previously described method [13]. The method, adapted for this study with minor modifications, can be briefly summarized as follows: 1 mL of various oils was placed into 5 mL screw-capped glass vials. Subsequently, increasing amounts of etodolac (10–250 mg) were added to the vials. The mixtures were then placed in an isothermal shaker (Wise Bath, Daihan Scientific Co., Ltd., Wonju, Republic of Korea) at 65 °C, then allowed to equilibrate for 24 h. Following each addition, a visual inspection was conducted at room temperature to determine the presence of drug crystals.

The selection of the surfactant was carried out based on a previously modified method [14]. In brief, the emulsifying properties of different surfactants were assessed by preparing coarse emulsions. A mixture comprising 5% (*w*/*w*) oil, 1% (*w*/*w*) surfactant, and a 10 g aqueous dispersion was prepared at a temperature of 80 ± 2 °C. The blending process involved using a high shear homogenizer (Silent Crusher M, Heidolph, Schwabach, Germany) operating at 25,000 rpm for 1 min. Visual observations revealed the presence of distinct emulsion characteristics when the mixture was cooled to room temperature.

#### 2.2.2. Critical Quality Attributes (CQAs) and Optimization

The optimization of the nanoemulsion composition was performed using a D-optimal mixture design with the assistance of the Design Expert Software (version 13.0.2.0, State-Ease Inc., Minneapolis, MN, USA). In this process, the independent variables of oil and surfactant were manipulated within the specified ranges of 5–8% and 1–5%, respectively. As demonstrated in previous studies, the determined critical quality attributes (CQAs) for nanoemulsions include parameters such as the droplet size (DS), polydispersity index (PDI), zeta potential (ZP), and encapsulation efficiency (EE) (shown Table 1) [14,15]. A rational approach called the Failure Mode, Effects, and Criticality Analysis (FMECA) was utilized to identify and record the parameters that are likely to affect the CQAs of the nanoemulsion [16]. The identified failure modes were prioritized based on risk priority numbers (RPNs), and it was found that the composition of the nanoemulsion, particularly the ratio of the oil and surfactant, had the highest RPN scores; therefore, the focus of the study was primarily on the nanoemulsion composition, considering the challenges associated with temperature control during the experimental procedures. A 13-run, mixture process variable design of experiments was conducted to examine these factors, as detailed in Table 2. It is worth noting that all 13 nanoemulsions were analyzed based on the phase specifications listed in Table 1.

#### 2.2.3. Preparation of Nanoemulsions

Nanoemulsions were produced using both high shear homogenization and ultrasonication methods. Initially, coarse emulsions were prepared by dispersing oil in an aqueous surfactant solution using a high shear homogenizer (SilentCrusher M, Heidolph, Germany) at 20,000 rpm for 1 min. Subsequently, probe ultrasonication (Bandelin, Sonoplus HD 2070, Germany) was employed to reduce the size of the coarse emulsions. The process parameters were determined based on a previously reported method [15], with no significant differences observed between the application durations and amplitude levels; therefore, the shortest duration of 5 min and an amplitude of 75% were chosen for the formulation preparation.

#### 2.2.4. Quantification of Etodolac

A quantification study was performed using a high-pressure liquid chromatography (HPLC) instrument (Agilent 1100, Santa Clara, CA, USA). For the analysis, a previously established analytical method was employed with minor adjustments [17]. The mobile phase, composed of acetonitrile, water, and glacial acetic acid (in a ratio of 61:38:1, *v*:*v*:*v*), filtered through a 0.45μm membrane filter, sonicated prior to use. The separation was carried out using a 4.6 × 250 mm long column with a particle size of 5 μm (Thermo Scientific, BDS Hypersil, C18, Waltham, MA, USA) at 40 ± 0.5 °C. The analysis was performed by isocratic elution at a flow rate of 1.2 mL/min, while the sample injection volume was 40 μL, and etodolac was detected at 280 nm. The retention time of etodolac was 4.53 min. Data collection was carried out using the Agilent ChemStation^®^ Software (Rev. B. 04.03-SP2 (105)). Standard validation procedures were followed to validate the method before conducting any experiments [18].

#### 2.2.5. Characterization Studies

##### Determination of Droplet Size, Polydispersity Index and Zeta Potential

The droplet size (DS) and polydispersity index (PDI) of the formulations were determined using a dynamic light scattering (DLS) instrument (Zetasizer Nano ZS, Malvern, PA, USA). The samples were diluted (1:20) and placed in a cuvette for measurement. All measurements were conducted at room temperature. The measurement times and numbers were adjusted automatically. Additionally, for the determination of the zeta potential (ZP), the instrument was operated in the zeta mode, and the measurements were performed at 25 °C using a capillary electrode cuvette instead of a regular cuvette [14].

##### Morphological Analysis

Scanning electron microscopy (SEM) was employed to observe the droplets of nanoemulsions. As previously described, diluted samples with water (1:20, *v*/*v*) were placed on a glass slide and left to dry in a vacuum oven at room temperature [14]. The surfaces of the dried nanoemulsions were coated with a layer of gold and palladium using a sputter coating technique (Bal-tec, Lübeck, Germany). The coated samples were then examined using a SEM (Zeiss EVO 40, Dublin, CA, USA). Micrographs of the nanoemulsions were captured at various magnifications using the instrument.

##### Measurement of Encapsulation Efficiency (EE) and Loading Capacity (LC)

A previously described method was adapted to determine the encapsulation efficiency (EE) and loading capacity (LC) [14]. Briefly, an aliquot of formulation was taken and centrifuged at 5000 rpm for 20 min. In order to determine the free drug, the supernatant was quantified with HPLC. The EE and LC were calculated according to the following Equations (1) and (2):(1)$$\mathrm{EE}=\frac{\mathrm{initial}\;\mathrm{amount}\;\mathrm{of}\;\mathrm{etodolac}-\mathrm{free}\;\mathrm{etodolac}}{\mathrm{initial}\;\mathrm{amount}\;\mathrm{of}\;\mathrm{etodolac}}\times 100$$
(2)$$\mathrm{LC}=\frac{\mathrm{initial}\;\mathrm{amount}\;\mathrm{of}\;\mathrm{etodolac}-\mathrm{free}\;\mathrm{etodolac}}{\mathrm{amount}\;\mathrm{of}\;\mathrm{lipid}\;\mathrm{in}\;\mathrm{the}\;\mathrm{formulation}}\times 100$$

##### Fourier Transform Infrared (FTIR) Analysis

Potential interactions among the components of the formulation were examined using a Fourier transform infrared (FTIR) spectrometer (NICOLET iS50, Thermo Scientific, USA). As previously reported, the active substance, as well as the medicated and unmedicated nanoemulsions (lyophilized), were separately analyzed in the wavelength range of 4000–400 cm^−1^ using the transmission mode of the FTIR instrument equipped with the Omnic version 9.0.0 software [15]. Multiple scans were performed for each sample, and the applied force over the sample was adjusted to obtain satisfactory transmittance results.

##### Measurement of Viscosity and pH

The viscosities of the preparations were determined by introducing them into the cuvette of a viscometer and employing a spindle S-7 (DV-II viscometer, Brookfield, WI, USA) with a shear rate of 100 cm^−1^ at ambient temperature. The pH values of the preparations were measured using a digital pH meter (MP220, Mettler Toledo, Hessen, Germany).

##### Stability Study

A stability study was conducted according to a previously established method [19]. Briefly, the formulations were stored in a controlled humidity chamber (Binder, Germany) at different temperatures (25 °C and 40 °C) for 90 days. Samples were taken regularly at specified intervals to monitor changes in their DS, PDI, ZP, EE, pH and viscosity.

##### In Vitro Drug Release Study

The release characteristics of the formulations were evaluated through in vitro experiments using Franz diffusion cells (Permegear, Hellertown, PA, USA). Each cell had a diffusion area of 2.88 cm^2^, and the receptor chamber had a volume of 20 mL. The release study was conducted under sink conditions, with 2 g of the formulations introduced into the donor chamber. The donor chamber was separated from the receiver chamber by an artificial membrane (Spectra/Por Dialysis membrane, Thermo Scientific, Waltham, MA, USA) with a molecular weight cut-off of 3.5 kDa. At designated time intervals, 0.7 mL samples were collected from the release medium (PBS, pH 7.4), and the volume was replenished with an equal amount of fresh medium. The samples were analyzed using the validated HPLC method. The assessment of the in vitro release patterns was evaluated through the application of various models, namely, the zero-order, first-order, Higuchi, Hixon–Crowell, and Korsmeyer–Peppas models, as outlined in Equations (3)–(7), respectively [20]:Q_t_ = Q_0_ + k_0_t (3)
Q_t_ = Q_∞_ (1 − e^−k_1_^^t^) (4)
Q_t_ = Q_0_ + k_H_t^1/2^
(5)
W_0_ ^1/3^ − W_t_ ^1/3^ = Kt (6)
log [Q_t_/Q_∞_] = log k + nlogt (7)
where Q_t_ and Q_0_ represent the drug quantities released at a specific time (t) and at the initial time (t = 0) into the release medium, respectively. k_0_, k_1_, and k_H_ represent the release rate constants for the zero-order, first-order, and Higuchi models, respectively. W_0_ and W_t_ denote the drug amounts in the formulation at the beginning and at a given time (t), respectively. K is a constant that encompasses the surface–volume relationship. Q_t_/Q_∞_ signifies the fractional drug release. k and n represent the kinetic constant and diffusional release exponent, respectively.

##### Ex Vivo Permeation Study

An ex vivo permeation study of nanoemulsions was carried out on porcine skin using Franz diffusion cells. The parameters of the permeation study were adapted from the previous study [17]. Briefly, the hair was eliminated using an electric trimmer and abdominal skin was carefully excised. The fat and particles attached to the skin specimens were gently eliminated and preserved at −25 °C for subsequent use in diffusion studies. Prior to commencing the diffusion experiments, the skin samples were immersed in an isotonic saline solution for a duration of 30 min. Afterwards, the skin samples were positioned in Franz diffusion cells, with the dermal side in contact with the receptor medium and the epidermal side in contact with the donor chamber. The donor chamber of the Franz cells was loaded with two grams of the formulation. Additionally, a control group consisting of a coarse suspension of etodolac (1%, *w*/*w*) in a carboxymethyl cellulose (CMC) dispersion (0.5%, *w*/*w*) was also prepared. The receptor phase, consisting of phosphate-buffered saline (PBS) at pH 7.4, was maintained at a temperature of 37 ± 0.5 °C with continuous stirring at 160 rpm. At specific time points over a 24 h period, 0.7 mL samples were collected from the receptor phase and replaced with an equal volume of fresh PBS under the same temperature and stirring conditions. The permeation study was conducted under sink conditions. The collected samples were then analyzed using a validated HPLC method. The coefficient constant, steady state flux (Jss), and Kp (permeability coefficient) values were determined based on the findings of the ex vivo permeation study.

Previously established tape stripping methods were adapted to quantify the amount of etodolac in the Stratum corneum [19,21]. Following the completion of the ex vivo study, the skins were retrieved from the receptor chamber of the Franz cell. Any excess formulation present on the skin was gently eliminated using cotton buds and washed with PBS. After washing the skins in PBS to remove any excess formulation, regular sellotape (VeGe^®^, Istanbul, Turkey) pieces were applied to the skin surface with light pressure and then removed. The initial two tape applications were excluded as they primarily capture formulation residue on the skin surface. Fifteen consecutive tape strips were sequentially used to separate the Stratum corneum from the epidermis and dermis. These tape strips were collected and placed in tubes containing a mixture of 10 mL acetonitrile and PBS (75:25, *v*/*v*). Then, they were extracted for 24 h. The tape strips were extracted for 24 h in a shaker, and then the strips were removed. The extracted solution was evaporated using a rotary evaporator (Heidolph, Germany), and the resulting residue was reconstituted with 2 mL of PBS. The sample was vortexed for 5 min and subsequently transferred to HPLC for quantification.

After the tape stripping study, the utilized skin sections were divided into smaller segments. Subsequently, they were transferred to homogenization tubes containing 5 mL of a mixture of acetonitrile and PBS in a ratio of 75:25 (*v*/*v*). The mixture was homogenized using a tissue homogenizer (Silent Crusher S, Heidolph, Germany) at a speed of 10,000 rpm for 5 min. Following homogenization, the preparation was vortexed for 1 min, sonicated for 20 min, and then centrifuged at 3000 rpm for 5 min. The resulting preparation was filtered through a 0.45 µm pore membrane, and the filtrate was analyzed using HPLC to determine the concentration of etodolac.

#### 2.2.6. Preparation of Etodolac Nanoemulsion-Based Gel

A secondary carrier system was needed to provide easy and effective topical delivery of etodolac-loaded nanoemulsions. Previously, carboxymethyl cellulose (CMC) was utilized for the topical delivery of etodolac for a penetration enhancement study [17]. Thus, the lowest concentration of CMC was selected (0.5%, *w*/*w*) for the application consistency and no or the lowest interference with the nanoemulsion’s diffusivity. Moreover, CMC was reported as a biocompatible and bioadhesive agent; thus, CMC could be considered as a perfect candidate for providing the desired consistency [22,23]. Briefly, a 0.5 g amount of CMC was added to the 99.5 g of etodolac-loaded nanoemulsion dispersion. This etodolac nanoemulsion-based gel system was abbreviated as ETD-NE-CMC for the subsequent studies. The formulations were kept at room temperature in a dark place and remained so over a night before the studies to remove entrapped air.

#### 2.2.7. In Vivo Study

##### Animals

The in vivo experimental protocol received ethical approval from the Institutional Animal Care and Use Committee of Yeditepe University, with the corresponding protocol number 2021–043 and decision number 2021–08/6. The study strictly adhered to the ethical guidelines outlined by the European community for the welfare and treatment of animals [24]. A total of twenty-four male Wistar albino rats weighing between 240 and 250 g were sourced from the Yeditepe University Animal Reproduction Center (YUDETAM) in Istanbul, Turkey. The rats were kept in a controlled environment with standard temperature and relative humidity conditions. The rats were randomly assigned to four groups, with each group consisting of six subjects. Throughout the study, the rats were provided with a standard diet and unrestricted access to water. Daily observations were made to record various parameters, including weight, food, and water intake.

##### Induction of Inflammation

The anti-inflammatory activity of the preparations was assessed using the carrageenan-induced hind paw edema model in rats [7]. The rats were randomly divided into four groups (n = 6) including a control group, placebo group, conventional delivery group, and novel delivery group. Carrageenan (λ type 1, Santa Cruz, Dallas, TX, USA) was dissolved in a saline solution. The preparations, containing a dose of 20 mg/kg of etodolac, and the placebo gel were topically applied to the plantar surface of the right hind paw of the rats and gently rubbed. After 30 min, each rat received an intraplantar injection of 100 µL of a 1% *w/v* carrageenan solution into the right hind paw, and 100 µL of physiological saline was injected sub-plantarly into the left hind paw. The volumes of the right and left paws were measured using a digital plethysmometer (Panlab Harvard Apparatus, Barcelona, Spain) before and at 1, 3, 4, 6, and 24 h after the carrageenan injection. The difference in volume between the right and left paws was considered as the paw edema volume. The percentage inhibition of edema was calculated as previously reported [7]:(8)$$Inhibition\;of\;edema=\left(1-\left(\frac{Vt}{Vc}\right)\right)\times 100$$
where *Vt*: the paw edema volume of the groups treated with the formulations in rats, and *Vc*: the mean edema volume of the control group.

##### In Vivo Experimental Group Design

After successfully inducing inflammation in the rats, they were divided into various groups based on the following criteria:

Group 1: control group (animals with induced inflammation but who received no treatment, n = 6),

Group 2: placebo group (animals with induced inflammation treated with an unmedicated nanoemulsion-loaded CMC gel, n = 6),

Group 3: conventional group (animals with induced inflammation treated with a coarse powder of etodolac (20 mg/kg dose of etodolac) dispersed in a CMC gel, n = 6),

Group 4: novel group (animals with induced inflammation treated with an etodolac-loaded nanoemulsion (20 mg/kg dose of etodolac) dispersed in a CMC gel, n = 6).

#### 2.2.8. Statistical Analysis

Edema scores obtained in the in vivo studies were evaluated statistically with the Newman–Keuls multiple comparison test following a one-way analysis of variance (ANOVA), and the scoring value was statistically evaluated with the Chi-square test. Conversely, a one-way ANOVA test and Student *t*-test comparison test were also used to evaluate the release, diffusion, and bioavailability parameters of the etodolac, while α = 0.05 (*p* ≤ 0.05) was accepted as the degree of significance in both separate studies.

## 3. Results and Discussion

### 3.1. Selection of Excipients

As an integral part of pre-formulation studies, the selection of the oil and surfactant is crucial. Previously established and adapted methods were utilized to determine the oil and surfactant types [13,14]. An oil screening study was conducted using various oils to assess the compatibility and solubility of etodolac. Isopropyl isostearate (IPIS) was identified as the oil in which etodolac exhibits the highest solubility (Appendix A).

Regarding the surfactant selection, non-ionic surfactants (Poloxamer^®^ 188, Poloxamer^®^ 407, Brij^®^ 35, tyloxapol, and Tego Care^®^ 450) were screened due to their low cytotoxicity and high biocompatibility. They are generally recognized as safe (GRAS) substances. Non-ionic surfactants are minimally affected by changes in ionic strength and pH. Several non-ionic surfactants were assessed to observe their emulsifying properties. Based on the visual outcomes of the screening study, Tego Care^®^ 450 provided non-viscous and phase-separation-free nanoemulsions (Appendix A).

### 3.2. Optimization of Nanoemulsions

Prior to the optimization study, the FMECA methodology was used to assess potential failure modes, effects, and criticalities. It is a structured approach used to identify and record parameters that are likely to affect the critical quality attributes (CQAs) of the nanoemulsion [16]. In the preliminary studies, the consideration of the process parameters (e.g., homogenization rate and duration, sonication amplitude and duration, and temperature) and the formulation’s composition (including the API, surfactant, oil, and water ratio, etc.) was comprehensively discussed and scored. The process parameters demonstrated a lower risk priority number (RPN) compared to the formulation composition. Consequently, these identified CQAs were selected as the responses based on the formulation composition (highest RPN value).

An initial optimization study was carried out using a D-optimal mixture design to determine the appropriate amounts of excipients for the formulations (Figure 1). In this study, the independent variables were identified as the oil and surfactant, ranging from 5% to 8% (IPIS) and from 1% to 5% (Tego Care^®^ 450), respectively. The correlation coefficients (R2) obtained for the DS, PDI, and ZP after designing the nanoemulsion formulations were 0.988, 0.954, and 0.976, respectively (Appendix A). The design of the placebo nanoemulsions demonstrated compatibility with the linear model. 

Based on the results of the initial optimization study (F1–F3 and F9–F11), the surfactant ratio in the formulations was inversely proportional to the DS and PDI. The reduction in the DS and PDI of the placebo nanoemulsions with an increase in the emulsifier concentration can be explained based on their surface activity. As the emulsifier concentration increased, its effect on the surface tension of the system also escalated. Energy conditions became favorable for the maximum partitioning of the oil phase, consequently facilitating optimal contact between the oil and aqueous phases. In a study, the impact of surfactant concentrations on droplet formation was thoroughly investigated between 0.5% and 5% (*w*/*w*) [25]. In general, a comparable trend was noted in that study; however, the process parameters posed constraints on reducing the droplet size, hence, there was an observed need for optimization to prevent the excessive use of surfactants.

The dependent variables were the DS, PDI, and ZP. After the oil and surfactant ratio of the appropriate formulation was obtained with the 5^th^ formulation, a second formulation design was performed to determine the amount of active substance (etodolac) ratio to be loaded (Table 2). For the management of chronic pain associated with arthritis, the recommended dosage of etodolac falls within the range of 400 mg/day to 1200 mg/day, administered in divided doses. Furthermore, in cases necessitating acute pain management for arthritis, frequent dosing within the range of 200 mg to 400 mg is advised [26]. The lowest reported topical dose of etodolac stood at 1% (*w*/*v*) (*w*/*v*) [17], while the highest amount was set at 5% (*w*/*v*) in accordance with prior research [26]. Consequently, a secondary experimental design was conducted within this range of ratios (Figure 2). The software generated a total of nine experimental runs, and the design matrix detailing these runs is provided in Table 3.

Out of all the formulations tested, only one (Formulation 2) met the CQA specifications according to Table 3. As a result, numerical optimization was used to identify the optimum formulation, which had a better oil composition after achieving a notably effective percentage of etodolac amount. The desirability factor (Df) was obtained as 0.9877 for the formulation that contains 8% oil, 5% surfactant, and 1% etodolac (F2); thus, further characterization studies were conducted on the F2-coded nanoemulsions.

In the context of nanoemulsions, the PDI value, along with the DS, is regarded as one of the CQAs. Ideally, this value is anticipated to fall within a narrow range, typically <0.2 [3,4]. Nanoemulsions undergo a phenomenon referred to as Ostwald maturation, where smaller droplets decrease in size while larger droplets grow. This phenomenon occurs because of the diffusion of oil molecules between droplets within the continuous phase. The process is unequivocally driven by the Kelvin effect, which stipulates that smaller droplets within the emulsion exhibit a higher oil solubility in comparison to larger droplets, primarily due to differences in the Laplace pressures. Our comprehensive study undeniably affirms that the most optimal droplet size distribution is attained by employing a high surfactant concentration (5%) and a low ETD concentration (1%) for the production of nanoemulsions [27].

In the present study, the ZP was assigned as one of the CQAs because it plays a pivotal role in guaranteeing the physical stability of nanoemulsions. When the emulsifier system used has a low molecular weight and a ZP of more than ±30 mV, the stability is good and can become excellent when the ZP reaches ±60 mV [3,4]. The formulations demonstrate ZP values within the range of −30 mV to −42 mV, as indicated in Table 2 and Table 3. This range signifies that nanoemulsions can uphold their physical stability over an extended duration.

In the current optimization study, the EE was also designated as a CQA. The EE is expressed as the proportion of the drug present in the nanoemulsion batches relative to the total amount of drug used. The experimental errors for the batches employed in the optimal design are detailed in Table 3. The nanoemulsion batches exhibited a range of EE values, ranging from 72.46% to 92.3%. Notably, a formulation comprising 1% etodolac, 8% isopropyl isostearate, and 5% Tego Care^®^ 450 yielded a notably higher percentage EE of 92.46%.

### 3.3. Quantification of Etodolac and Analytical Method Validation

Ensuring accurate and comprehensible information regarding a sample requires the initial confirmation of the analysis method. Establishing a consistent and reliable analytical strategy is the initial step in the ongoing process of method validation during development. Consequently, after the successful production of an etodolac-loaded nanoemulsion, HPLC was employed to collect the chromatography data. The obtained chromatograms confirm the absence of interference between the excipient peak and the etodolac peak, providing a reliable and separate identification of both components. The retention time of etodolac was determined to be 4.53 min (Appendix A). The analytical method exhibited a linear relationship within the concentration range of 0.05 to 5 µg/mL, as indicated by the regression equation y = 0.2292x + 0.824, with a high coefficient of determination (R^2^ = 0.9993). The accuracy of the method, as well as its intra-day and inter-day precision were evaluated, and the relative standard deviations (RSDs) were determined to be below 2%. This indicates that the method exhibits a high precision and accuracy in measuring the target compound. The recovery of etodolac was found as 98.55 ± 0.26%–101.78 ± 1.53%. Limits of detection (LOD) and quantification (LOQ) were calculated as 5.28 ng/mL and 15.26 ng/mL, respectively.

### 3.4. Morphological Evaluation

Surface morphology of the optimal formulation was revealed by using SEM. The micrographs of nano droplets are presented in Figure 3. The images indicate that the nanoemulsions were predominantly spherical and free from the etodolac crystals. The droplet size measurements of the images have similar outcomes with the DLS findings. The literature data also supports the surface morphology of nanoemulsions [14].

### 3.5. FTIR Analysis

The FTIR spectra depicted in Figure 4 illustrate the FTIR spectrum of etodolac, etodolac-loaded nanoemulsions, and medicated nanoemulsion-loaded gels. Within the FTIR spectrum of etodolac, the N-H stretching vibrations originating from the secondary amine group were identified at 3338 cm^−1^. The C-H stretching vibrations are described in the region 3060–2810 cm^−1^ [28]. Peaks at 2850 cm^−1^, 2930 cm^−1^, 2986 cm^−1^, and 3026 cm^−1^ were associated with the aromatic C-H stretching in the FTIR spectrum of etodolac. The carbonyl C=O vibration was anticipated within the 1750–1600 cm^−1^ range [28]. Specifically, the peak at 1736 cm^−1^ was attributed to the carbonyl group of carboxylic acid stretching in the FTIR spectrum of etodolac. Peaks within the 1500–1400 cm^−1^ range were construed as aromatic ring C-C stretching vibrations, consistent with prior data [28]. This analysis serves to chemically confirm the structure of pure etodolac.

In both the spectra of medicated nanoemulsions and medicated emulsion-loaded gels, the distinctive peaks associated with etodolac are readily discernible. Notably, the N-H stretching vibrations at approximately 3340 cm^−1^, the carbonyl vibrations within the 1750–1600 cm^−1^ range, and the aromatic C-H stretching in the 3060–2810 cm^−1^ range were all observed to remain intact. The intensity of the distinctive peaks associated with etodolac exhibited a remarkable decrease in the spectrum of etodolac-loaded nanoemulsions when compared to pure etodolac. Peaks in the C-H stretching region (3060–2810 cm^−1^) were found to be flattened and a reduced intensity was observed for the carbonyl peak at 1736 cm^−1^; thus, it is reasonable to conclude that the morphology of the etodolac changed. A scientific report similarly interpreted this alteration as an amorphization or complexation of etodolac [29]. In this case, it could be interpreted as an efficient encapsulation of etodolac into the oil droplets.

### 3.6. In Vitro and Ex Vivo Release Studies

Prior to the in vitro drug release studies, etodolac-loaded nanoemulsions were prepared as aqueous dispersions. Then, carboxymethyl cellulose (CMC) was added to formulate a gel system as a secondary carrier; thus, the in vitro release study was performed on the final preparation. Considering the reported solubilities of etodolac (aqueous solubility of etodolac: 30.2 µg/mL), the release medium and its volume were adjusted to maintain the sink conditions [30]. 

The in vitro release study of the drug was performed using a Franz diffusion cell, where the receptor medium consisted of a phosphate buffer (20 mL, 0.05 M, and pH 7.4). The experimental conditions and key parameters for these studies were based on previous reports [7,17]. The release profiles are presented in Figure 5. The most intense etodolac diffusion from the preparations occurred within the initial eight hours. Consequently, the drug’s penetration during the early hours was deemed crucial for establishing the permeation profile of the gel formulations. Moreover, it is impractical to maintain gels on the skin surface for an extended period. Eight hours represents a reasonable duration for the persistence of a semi-solid formulation on the skin surface and is suitable for mimicking the actual contact time of gels with the skin. Additionally, it is worth noting that the skin can act as a reservoir for the active substance, releasing it even after the formulation has been removed; therefore, this research primarily focused on the initial eight hours of the release profiles.

In the first hour of the in vitro release study, an initial burst effect was observed from the nanoemulsion-loaded gel (ETD-NE-CMC), after that, the release profile followed the sustained release pattern. This result has been found to be in accordance with the literature data and is considered to be due to the adsorption of etodolac on the droplet’s surface or its dispersion into the surfactants [14,31]. Compared to coarse dispersion (control group), a significant difference was observed within the first hour (*p* < 0.05), with the coarse dispersion displaying a controlled release pattern. This difference may have originated from the particle size and surface area of the drug substance.

In a mathematical perspective, the release patterns of the coarse dispersion (control group) and nanoemulsion-loaded gel (ETD-NE-CMC) were fitted to zero order and Korsmeyer Peppas kinetic models, respectively (Table 4). In zero order kinetics, the release of an active agent solely depends on time, and the process occurs at a consistent rate regardless of the concentration of the active agent. Considering the definition, the zero-order release pattern was also observed, as seen in Figure 5a. The release kinetic of the nanoformulation was fitted to the Korsmeyer Peppas model with a 0.769 release exponent (n), which means a non-Fickian transport. A range of 0.5 < n < 1 is reported as a non-Fickian or anomalous transport, and the mechanism of drug release is governed by diffusion and swelling [32]. In nanoemulsion delivery systems, this condition could be fitted to a diffusion-type release mechanism. The literature data for the etodolac nanoformulations, interestingly, demonstrate similar findings. In a study, nanostructured lipid carriers of etodolac exhibited an n value in the range from 0.558 to 0.749, which reveals the release mechanism is based on the drug diffusion through the lipid matrix [9]. In another study, cubosomes of etodolac displayed “n” values of 0.61 to 0.82, that reveal that the mechanism of drug release is governed by diffusion [33].

Several methods are available for the determination of skin integrity. Trans epidermal electrical resistance (TEER), trans epidermal water loss (TEWL), and trans epidermal water flux (TWF) were reported as standard methods [34]. The limit values for the TEER, TEWL and TWF were 1 kΩ, 10 g m^−2^ h^−1^ and 2.5 × 10^−3^ cm h^−1^, respectively. Various commercial instruments are presently accessible for measuring the TEWL. These instruments find primary applications in in vivo TEWL measurements, particularly in the domains of dermatological examinations and formulation development. Notably, a recent advancement includes the commercial release of an in vitro TEWL probe, facilitating initial assessments utilizing excised skin samples [35]. The absence of this type of probe in most research laboratories underscores a constraint of this study. On the other hand, alternative approaches such as visual microscopic examination offer researchers insights into skin integrity. Through a visual examination method adapted from a prior study, it was established that the skin integrity was found satisfactory [36].

Permeation profiles of the preparations are shown in Figure 5b. The determination of the permeability coefficient and flux were adapted from the previous study [7]. The permeability coefficient was determined by dividing the flux by the total donor concentration of the formulation. To calculate the flux (Jss) values, the cumulative amount of drug permeated in micrograms per square centimeter was graphed against the time in hours, with the slope representing the flux. As seen in Table 5, the flux and permeability coefficients of the dialysis membrane were significantly higher than the porcine skin (*p* < 0.05). This could be attributed to the intricate composition of the porcine skin, which resulted in increased resistance during the diffusion process for the drug molecule penetration [17]. Moreover, nanoemulsions exhibited a remarkable penetration enhancement in both the in vitro (dialysis membrane) and ex vivo (porcine skin) studies compared to the control group (Table 5). The permeation increase can be explained with the increasing saturation solubility, dissolution rate of etodolac by the means of nanosized droplets, the large surface area of the nanoemulsions, and the diffusion and concentration gradient. Similar findings were reported for etodolac nanosuspensions [7], etodolac-loaded solid lipid nanoparticles [8] and etodolac-loaded cubosomes [33].

Following the completion of the ex vivo study, the porcine skins were retrieved from the receptor chamber of the Franz cell. After the application of the technique described in the Material and Methods section, the percent permeation, percent drug amount in the *Stratum corneum* and the percent drug amount in other layers of the skin were calculated. The results are shown in Appendix A. The percent permeation of the nanoemulsion-loaded gel and coarse dispersion were 50.35 ± 0.46% and 19.0 ± 0.95%, respectively. The percentage of drug remaining in the *Stratum corneum* layer in the skins treated with the nanoemulsion-loaded gel and coarse dispersion was found to be 2.03 ± 0.08% and 65.01 ± 0.49%, respectively. The percentage of drug remaining in the other layers of the skins treated with the nanoemulsion-loaded gel and coarse dispersion was found to be 48.23 ± 0.16% and 15.27 ± 0.09%, respectively. Based on the results of the tape striping study, the Stratum corneum acted as a powerful barrier for the coarse powder of the etodolac dispersions. Moreover, the other layers also hindered the penetration of etodolac for both preparations. Despite all this, the nanoemulsions increased the penetration of etodolac through the skin by approximately 2.5 times compared to the coarse dispersion.

### 3.7. Stability Studies

The physical stability of the optimized final formulation was monitored in terms of the DS, PDI, ZP, EE, pH, and viscosity for 90 days at 25 °C (60% RH ± 5%), and 40 °C (75% RH ± 5%). Various time intervals (from day zero to 60 or 180 days) and test conditions (4 to 40 °C) for the assessment of the physical stability for nano delivery systems have been introduced in the literature [37,38]. A normal condition (25 °C (60% RH ± 5%)) and a forced condition (40 °C (75% RH ± 5%)) were selected for the stability test to observe the practical environmental effects. The stability outcomes are presented in Appendix A. 

There was no significant difference (*p* > 0.05) between the production day (day zero) and 90th day of stability for all physical stability parameters at 25 °C; however, there was a significant difference (*p* < 0.05) after the 30th day of stability for DS at 40 °C. Similarly, a significant difference (*p* < 0.05) was observed after the 30th day of stability for the viscosity at 40 °C. Moreover, after the 60th day, a significant difference (*p* < 0.05) was observed for the EE.

After the 30th day of the stability study, it was determined that there was a significant increase of the droplet size of the nanoemulsions (167.4 ± 1.3 nm to 193.9 ± 1.4 nm) at 40 °C. This situation could be elucidated by the occurrence of coalescence and the Ostwald ripening phenomenon. When coalescence is a potential degradation mechanism, the size distribution may exhibit various patterns, ranging from homogeneous to heterogeneous distributions (mostly heterogenous). Conversely, the Ostwald ripening process primarily governs the destabilization of nanoemulsions, whereas coalescence chiefly affects the destabilization of coarse emulsions. Temperature is a factor that accelerates the Ostwald ripening process. As shown in a previous study, nanoemulsions stored at 40 °C exhibit a faster Ostwald ripening mechanism compared to conditions at 25 °C [11].

ZP is the electrostatic potential on a nano droplet’s surface, which affects the physical stability, and skin and nano droplet interactions. The nanoemulsion displayed a negative electric charge of around −35 mV. This was attributed to the negatively charged carboxylic and hydroxyl groups of the sugar component in Tego Care^®^ 450 under neutral conditions. The substantial absolute value of the ZP provided stability to the nanoemulsions by inhibiting the droplet aggregation through a strong mutual repulsion. The literature states that a zeta potential of more than ±30 mV provides good stability and obtains an excellent stability when the ZP reaches toward ±60 mV [3]. The absence of a significant difference in the zeta potentials of the nanoemulsions stored at 25 °C and 40 °C can be explained by the high absolute ZP value.

After the 60th day of the stability study, it was determined that there was a remarkable decrease in the EE (%) of the nanoemulsion (92.69 ± 1.40% to 70.83 ± 1.59%) at 40 °C. The reason for this was thought to be owing to the leakage of the active substance due to the elevated diffusion rate at 40 °C. Similar destabilization cases were also reported for nano carriers [37,38]. 

The viscosity and pH are CQAs of the final formulation. The viscosity of topical formulations varies between 14 and 19 cP, and the pH value should be between 5 and 7. In both cases the final formulation met the criteria. After the 30th day of the stability study, the viscosity of the nano formulation significantly decreased (*p* < 0.05). Additionally, after the 60th day of the stability study, the pH of the nano formulation significantly decreased (*p* < 0.05). This condition could be a result of increasing the acidic character of the dispersion medium due to etodolac (due to carboxylic acid moiety) leakage from the droplets. Similar changes in both viscosity and pH were reported previously [37].

### 3.8. In Vivo Studies

Carrageenan-induced rat paw edema serves as a commonly employed experimental model for investigating in vivo anti-inflammatory effects [39]. The injection of carrageenan triggers an acute and localized inflammatory reaction characterized by distinct phases. The initial phase (≤1 h) results from the release of histamine, bradykinin, serotonin, and substance P, while the later phase (>1 h) is linked to the generation of pro-inflammatory mediators [7,39]. The rats in the control and placebo groups exhibited an elevation in edema volume, which became noticeable within one hour and reached their maximum point eight hours after induction (Figure 6). The excipients included in the formulation were found to have no significant impact on the edema volume when compared to the control group and the placebo group (*p* > 0.05). The medicated preparations significantly decreased the edema volume after the 8th hour (*p* < 0.05).

The oedema volume in the control group continued to rise until the 8th hour. Application of the medicated nanoemulsion-incorporated gel (ETD-CMC-NE) continued to suppress paw oedema (edema inhibition of the ETD-NE-CMC group: 13.4%, 36.5%, and 50.65% for the 6th, 8th, and 24th hours, respectively) formation. A similar trend was reported for nanoemulsions of ibuprofen during the 24 h of an anti-inflammatory activity study [40]. In another study, etodolac microemulsions were formulated for topical administration, resulting in an up to 66.8% inhibition of edema in an anti-inflammatory bioactivity study, demonstrating the efficacy of etodolac for dermal delivery [41]. When other nano systems were considered, etodolac nanosuspensions displayed a remarkable edema suppression for 24 h [7]. 

Exactly as stated in numerous earlier publications, drugs can penetrate the skin through multiple pathways, which encompass an intercellular pathway across the lipid bilayer, a transcellular pathway through keratin-rich corneocytes, and an alternative pathway through the hair follicles and sweat ducts [1,2,3,4]. The pores in the *Stratum corneum*, which could potentially serve as entry points to the deeper layers of the skin, are estimated to have sizes ranging from 20 to 200 nm. Furthermore, hydration-induced *Stratum corneum* expansion can increase the pore size, promoting an enhanced drug penetration. Additionally, penetration enhancers can modify *Stratum corneum* hydrocarbon chains and lipid components, further facilitating drug permeation. The nanoemulsions here increased the penetration of the etodolac due to the formulation’s physicochemical properties, such as a small droplet size, charge, and viscosity. The suppression of paw edema in this study can be explained through these physicochemical properties.

### 3.9. Comparison of Current Work to Previous Reports

The findings obtained in the current study were compared with similar studies in the literature. In the first case, a conventional hydrogel system was designed to improve the penetration of etodolac via a topical route [17]. In that study, sodium lauryl sulphate, ethanol, and polyethylene glycol 400 (PEG 400) were used to solubilize etodolac and penetration enhancers in a CMC gel system. In in vitro (artificial membrane) studies, there was no significant difference between the control group and the formulations; however, in ex vivo studies, a significant difference (approximately a two-fold increase in permeability) was observed only in a formulation containing anethol as a penetration enhancer. Compared with the current study, the etodolac-loaded nanoemulsions (without organic solvents and penetration enhancers) showed a significant difference in penetration in both in vitro and ex vivo studies (approximately a two-fold increase). It is considered that this situation was due to the small droplet sizes of the formulation.

In another study, etodolac-loaded solid lipid nanoparticles (SLN) were prepared and dispersed into a gel system formulated with carbopol 934 for in vitro, ex vivo release studies, and in vivo experiments [8]. In our study, similar investigations were conducted, and it was observed that the in vitro trend in the current study increased slightly (attributed to the smaller particle size); however, in those ex vivo studies, it was observed that the nanoemulsions exhibited similar release profiles to the previous comparable ones. The anti-inflammatory responses were paralleled in the current study, but it was noted that the in vivo responses of formulations containing SLNs were higher compared to the current study. It is considered that this was due to the SLN carrier system providing a prolonged release compared to nanoemulsions.

In another study involving the preparation of etodolac-loaded nanostructured lipid carriers (NLCs), a formulation design was carried out using a design of experiments (DOE) approach, mirroring the methodology applied in our research [9]. During the characterization phase, parameters such as the particle size, PDI, ZP, and EE (%) were intensively examined, aligning with the current investigative procedures. Comparative analysis revealed that the NLCs maintained relatively larger sizes, with the PDI values displaying a tendency to deviate from a unimodal distribution (PDI > 0.2). Furthermore, it was noted that the NLCs, exhibiting an approximately 80% EE, showcased a ZP value akin to our investigation (−30 mV). When comparing the ex vivo release profiles of the nanoemulsions in the present study with those of the NLCs, a notable difference was observed, specifically, the NLCs exhibited a slower ex vivo release rate. This behavior is conjectured to stem from the presence of a solid matrix within the NLCs. 

In a study where etodolac nanosuspensions were prepared, the etodolac particle size was reduced to approximately 500 nm using physical methods, yielding a PDI value of less than 0.2 [7]. Similarly, a topical gel system was prepared using hydroxyethyl cellulose (HEC) and hydroxypropyl methyl cellulose (HPMC) as gelling agents, following a comparable approach with the current study. In vitro, ex vivo, and in vivo studies were conducted. When comparing the ex vivo permeation rates of nanosuspensions with the nanoemulsions in the current study, the permeation rate of the nanosuspensions was slightly lower. This is attributed to the droplet size and the flexibility of nano droplets. Upon examining the in vivo anti-inflammatory profiles, it was observed that both studies exhibited a similar trend in paw edema volume reduction. 

## 4. Conclusions

In the current investigation, nanoemulsions loaded with etodolac were formulated and optimized using a DOE approach. The primary objectives were to alter the route of administration and improve topical drug delivery. The characterization of the optimized formulation aligned with the CQAs established for the study. In vitro release and ex vivo permeation studies showed that nanoemulsion-loaded CMC gels are superior in terms of improving the penetration of etodolac. Furthermore, the final formulation exhibited stability for a duration of three months, which was attributed to its high ZP. Notably, the medicated nanoemulsion-loaded gels demonstrated a remarkable and enhanced anti-inflammatory activity when compared to the coarse dispersion of etodolac. Consequently, etodolac nanoemulsion-loaded gels present a promising and effective alternative delivery approach for the treatment of inflammation.

## Figures and Tables

**Figure 1 pharmaceutics-15-02510-f001:**
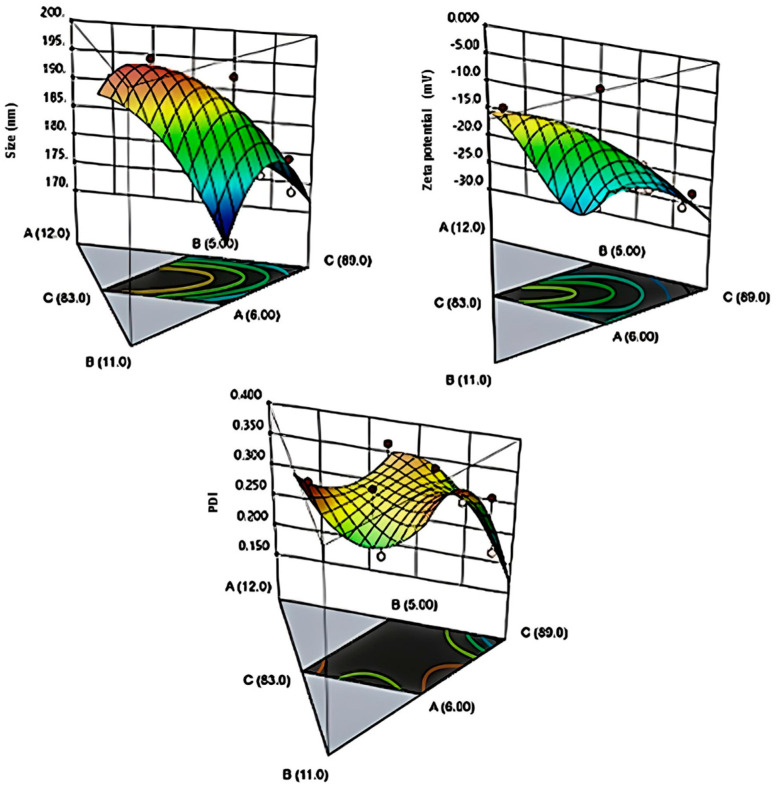
3D surface plots (D-optimal mixture design model) of the unmedicated nanoemulsion formulations generated by the Design Expert Software (version 13.0.2.0). (A: surfactant, B: oil, C: distilled water).

**Figure 2 pharmaceutics-15-02510-f002:**
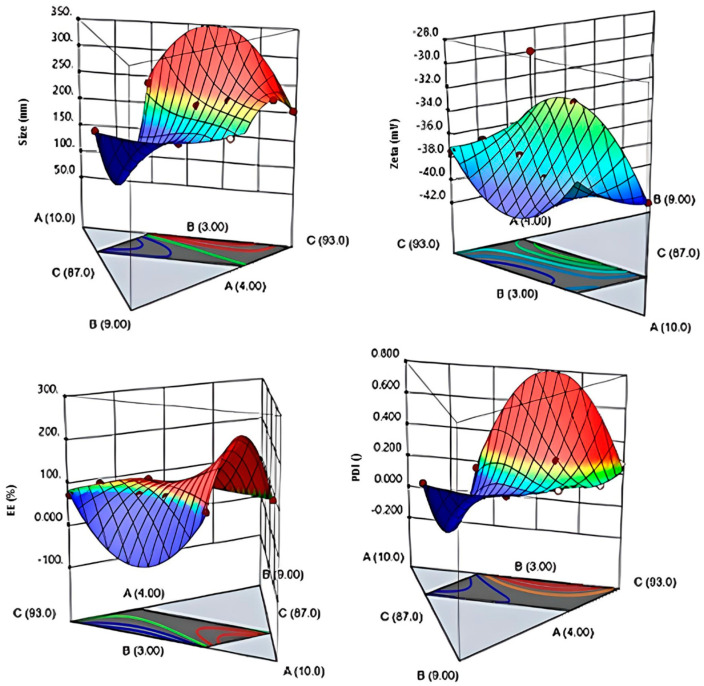
3D surface plots (D-optimal mixture design model) of the medicated nanoemulsion formulations generated by the Design Expert Software (version 13.0.2.0). (A: surfactant, B: oil, C: distilled water).

**Figure 3 pharmaceutics-15-02510-f003:**
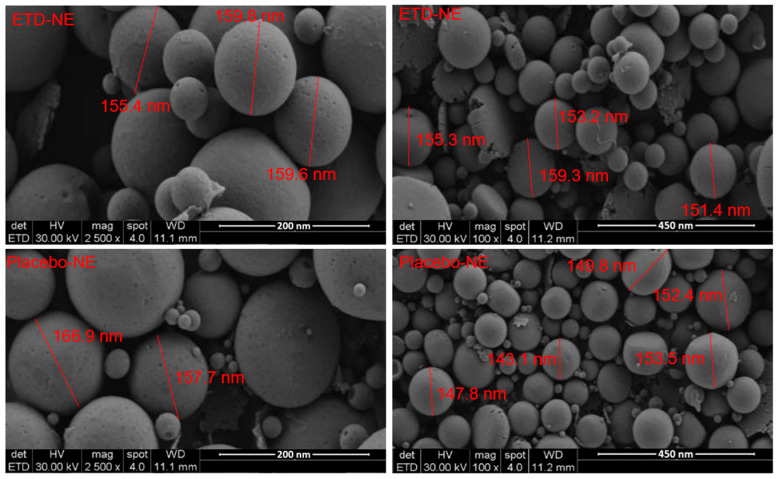
SEM images of medicated and unmedicated nanoemulsions (optimal formulation). The images on the right depict 2500× magnification; the images on the left depict 100× magnification.

**Figure 4 pharmaceutics-15-02510-f004:**
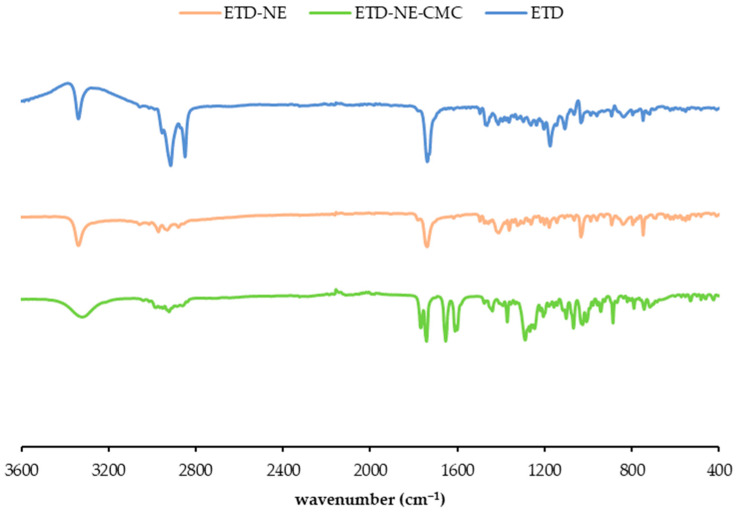
FTIR spectrum of etodolac and optimal formulations (ETD: etodolac, ETD-NE: etodolac loaded nanoemulsions, ETD-NE-CMC: medicated nanoemulsion-loaded gel).

**Figure 5 pharmaceutics-15-02510-f005:**
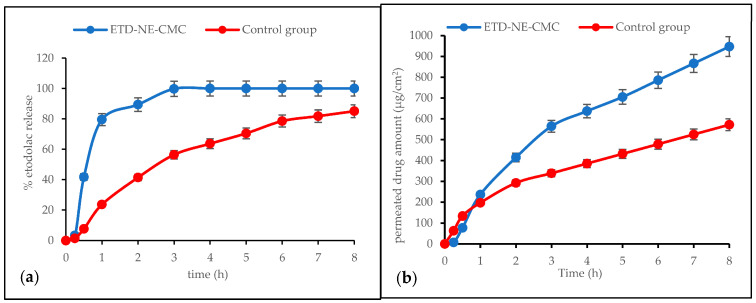
(**a**) In vitro cumulative release profiles of nanoemulsion-loaded gel (ETD-NE-CMC) and control group (n = 6). (**b**) Ex vivo permeation profile of the etodolac from the preparations through the skin (n = 3). The results represent a mean ± SD.

**Figure 6 pharmaceutics-15-02510-f006:**
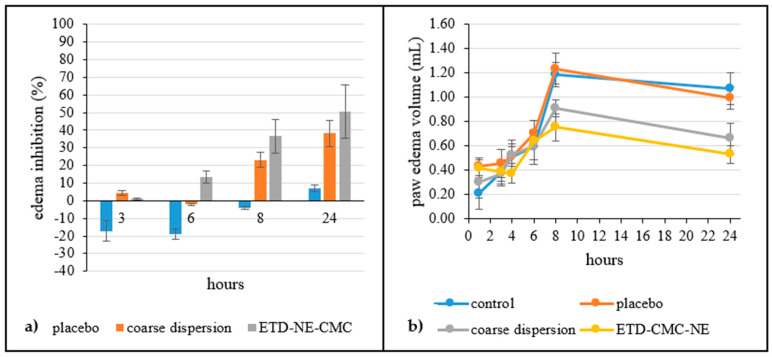
Percent edema inhibition of preparations (**a**) and paw edema volume (mL) (**b**) in rats (n = 6). The results represent a mean ± SD.

**Table 1 pharmaceutics-15-02510-t001:** CQAs, specifications, and brief testing method descriptions.

**Phase 1—Assessment of all generated formulations**	
DS (nm)	120–180 nm
PDI	≤0.2
ZP (mV)	≥|−30| mV
EE (%)	≥90%
**Phase 2—Assessment of selected formulation (90 day stability study)**	
DS (nm)	Droplet size change 20 nm (<10%)
PDI	≤0.2 (<10%)
ZP (mV)	≥|−30| mV (<10%)
EE (%)	≥90% (<10%)
pH	6.5–7.5 (<10%)
Viscosity (cP)	15.5–17.5 cP (<10%)

DS: droplet size, PDI: polydispersity index, ZP: zeta potential, EE: encapsulation efficiency.

**Table 2 pharmaceutics-15-02510-t002:** The design matrix contains formulation variables and responses that vary based on the concentration of oil used.

Formulation	Oil (%; X1)	Surfactant (%; X2)	ZP (mV; Y1)	PDI (Y2)	DS (nm; Y3)
F1	8	1.5	−42.4 ± 2.4	0.233 ± 0.03	198.6 ± 2.6
F2	6.6	3.3	−39.4 ± 0.9	0.351 ± 0.04	249.8 ± 3.3
F3	8	3.1	−33.7 ± 0.6	0.150 ± 0.04	164.6 ± 1.9
F4	5	2.8	−34.4 ± 1.3	0.184 ± 0.01	158.9 ± 0.9
F5	8	5	−36.0 ± 1.6	0.176 ± 0.03	159.2 ± 1.2
F6	5.6	3.2	−36.7 ± 2.1	0.355 ± 0.04	224.7 ± 2.4
F7	5.3	5	−38.1 ± 0.7	0.407 ± 0.03	240.3 ± 3.1
F8	6.8	4.7	−39.2 ± 1.5	0.178 ± 0.02	171.6 ± 2.7
F9	5	1	−33.4 ± 1.8	0.239 ± 0.04	196.8 ± 1.3
F10	6.5	1	−34.7 ± 0.5	0.193 ± 0.01	188.7 ± 1.4
F11	5	4	−31.6 ± 2.7	0.182 ± 0.04	176.8 ± 1.5
F12	6.6	4.2	−31.8 ± 2.5	0.190 ± 0.02	173.8 ± 1.5
F13	6.5	1	−34.9 ± 1.1	0.249 ± 0.01	221.5 ± 2.9

**Table 3 pharmaceutics-15-02510-t003:** Formulation variables and responses of design matrix.

Formulation	ETD (%)	DS (nm)	PDI	ZP (mV)	EE (%)
F1	3.2	175.5 ± 1.4	0.176 ± 0.03	−40.1 ± 2.4	73.01 ± 1.92
F2	1	163.5 ± 2.2	0.141 ± 0.02	−33.1 ± 1.7	92.30 ± 1.05
F3	2	179.5 ± 0.7	0.204 ± 0.04	−39.4 ± 1.7	91.02 ± 0.95
F4	1.8	184.7 ± 0.9	0.218 ± 0.03	−29.6 ± 1.6	92.00 ± 1.33
F5	4	230.0 ± 1.8	0.315 ± 0.04	−37.5 ± 0.9	73.52 ± 3.47
F6	3	215.6 ± 2.6	0.291 ± 0.03	−38.7 ± 1.5	80.19 ± 2.58
F7	4.1	239.3 ± 3.5	0.167 ± 0.01	−37.2 ± 0.7	73.51 ± 1.46
F8	5	203.8 ± 2.7	0.264 ± 0.01	−37.4 ± 2.2	72.46 ± 0.76
F9	2.6	235.1 ± 2.8	0.184 ± 0.02	−36.9 ± 1.6	88.70 ± 2.33

**Table 4 pharmaceutics-15-02510-t004:** Mathematical models of release kinetics.

Formulation	Zero Order (R^2^)	First Order (R^2^)	Higuchi (R^2^)	Hixson Crowell (R^2^)	Korsmeyer Peppas (R^2^/n)
Control group	0.9975	0.8519	0.9666	0.7995	0.9554/0.201
ETD-NE-CMC	0.9340	0.9707	0.9865	0.9600	0.9914/0.769

n: release exponent.

**Table 5 pharmaceutics-15-02510-t005:** Permeability coefficients and flux of preparations (n = 3). The results represent a mean ± SD.

Formulation	Flux (Jss) (µg/cm^2^ h)	Permeability Coefficient (Kp) (cm/h)	Permeability Coefficient (LogKp)	Permeated Amount at 8 h (µg × cm^2^)
Dialysis membrane				
Control group	783.4 ± 56.2	0.041 ± 0.009	−1.03 ± 0.06	8793.6 ± 630.0
ETD-NE-CMC	1068.4 ± 71.5	0.069 ± 0.005	−1.99 ± 0.03	14,336.9 ± 613.1
Porcine skin				
Control group	59.7± 15.2	0.004 ± 0.001	−0.72 ± 0.09	45.4 ± 1449.3
ETD-NE-CMC	165.7 ± 11.7	0.011 ± 0.001	−2.55 ± 0.17	357.2 ± 18.7

## Data Availability

Data will be made available on request.

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
