# Peer review of "Nanoemulsions as a Promising Carrier for Topical Delivery of Etodolac: Formulation Development and Characterization"

_pharmaceutics, 2023, doi:10.3390/pharmaceutics15102510_

Round 1

Reviewer 1 Report

The research work “Nanoemulsions as a Promising Carrier for Transdermal Delivery of Etodolac: Formulation Development and Characterization” is as per the scope of the journal. The nanoemulsion approach is used widely for topical and transdermal delivery to improve permeation and effective delivery of drugs.  The etodolac nanoemulsions were prepared using high shear homogenization and ultrasonication methods and evaluated for droplet size (DS), polydispersity index (PDI), and zeta potential (ZP) along with antiinflammatory activity by carrageenan-induced paw edema model in rats.

The comments for work are as follows:

1.       Line 34: As per title its transdermal preparation and in abstract it's topical. Please be uniform.

2.       Please cite method/standards: 2.2.2. Critical Quality Attributes (CQAs)

3.       Table 2 and 3: There are no significant differences in batches selected for globule size, zeta and PDI. Why was such a narrow design selected?

4.       Please add a method for preparing NE-CMC Gel. Why was CMC selected as gelling agent and in what concertation? How it will affect diffusion behaviours?

5.       Table 2 and 3: Please add standard deviation if all are average values.

6.       Figure 1 and 2: Please add name of Tool/software used in figure caption.

7.       In 3.5. FTIR analysis, initial few blue picks of a drug is not there in emulsion preparations. Why?

Author Response

Dear Reviewer,

We sincerely appreciate your valuable comments and contributions. We have completed the requested revisions and are resubmitting the manuscript for your reconsideration. Additionally, we have highlighted the changes made for easy tracking and provided a comment and response chart to clarify the modifications. Thank you for dedicating your precious time for this review.

Hereby, I would like to present the revisions we have made:

Comment_1: Line 34: As per title its transdermal preparation and in abstract it's topical. Please be uniform.

Response_1: The authors appreciate the reviewer's feedback. The terminological discrepancy regarding transdermal versus topical administration has been resolved. In line with the localized delivery of the active substance, 'topical' has been deemed more appropriate for this study, ensuring uniformity in the terminology throughout the manuscript.

Comment_2: Please cite method/standards: 2.2.2. Critical Quality Attributes (CQAs)

Response_2: The authors would like to thank the reviewer's suggestion. Critical Quality Attributes (CQAs) have been carefully defined and relevant citations have been incorporated. Additionally, the FMECA approach has been appropriately cited.

Comment_3: Table 2 and 3: There are no significant differences in batches selected for globule size, zeta and PDI. Why was such a narrow design selected?

Response_3: 

Thank you for the reviewer's observation. In previous nanoemulsion studies, we explored various oil and surfactant ratios:

  • 1-10% oil and 1-5% surfactant (doi: https://doi.org/10.1016/j.jddst.2018.03.011), and
  • 4-8% oil and 3-5% surfactant (doi: https://doi.org/10.1208/s12249-023-02551-6).

Given these experiences, we opted for a 5-8% oil and 1-5% surfactant range to target the Critical Quality Attributes (CQAs) outlined in Table 1. Specifically, in Table 2, formulations F5 and F8 met the CQAs, guiding our choice of lower particle size (F5) for enhanced permeation. Similarly, for Table 3, formulation F2 aligned with the CQAs.

If the “narrow design” indicates CQAs (from table1) (responses): In a study (doi: https://doi.org/10.1016/j.jddst.2022.103272) indicating that the range of 120-250 nm is effective for the topical administration of drugs, this range has been narrowed down to 120-180 nm. The PDI levels reduced to 0.2 to obtain an unimodal size distribution.

We adapted our study, taking this work (doi: https://doi.org/10.1016/j.jddst.2022.103272) as an example, by adjusting the droplet size to the range of 120-180 nm as mentioned in the study and reducing the PDI value to 0.2, which was indicated as 0.4. A more uni-modal distribution was desired. For the Zeta potential value, we initially intended to set a range greater than 30 mV (35-40 mV). However, following your feedback, we revised it to ZP ≥ |-30| mV. Consequently, we opted for the narrowest droplet size distribution, which we believed would result in the maximum permeation, and chose the smallest droplet size.

Comment_4: Please add a method for preparing NE-CMC Gel. Why was CMC selected as gelling agent and in what concertation? How it will affect diffusion behaviours?

Response_4: The authors wish to express their appreciation for the reviewer's valuable comments. In response to the queries raised, an additional section has been included to provide further clarification.

Comment_5: Table 2 and 3: Please add standard deviation if all are average values.

Response_5: The authors would like to thank the reviewer’s comment. The standard deviations were inserted and highlighted.

Comment_6: Figure 1 and 2: Please add name of Tool/software used in figure caption

Response_6: The authors would like to thank the reviewer’s comment. The software name was added to the figure caption and highlighted.

Comment_7: In 3.5. FTIR analysis, initial few blue picks of a drug is not there in emulsion preparations. Why?

Response_7: The authors would like to thank the reviewer's contribution. A comprehensive explanation was added to the last paragraph of the relevant section, the revised part was highlighted.

Best regards,

Reviewer 2 Report

The manuscript entitled “Nanoemulsions as a Promising Carrier for Transdermal Delivery of Etodolac: Formulation Development and Characterization” reports a study on etodolac loaded nanoemulsions designed to improve drug skin permeation.

The manuscript is well organized and the results are properly discussed. Unfortunately, the aim of the study is unclear. Transdermal delivery entails the achievement of drug therapeutic concentration in the blood after drug application on the skin surface. In the abstract, the authors concluded that the investigated “nanoemulsions are promising carriers for topical delivery of etodolac.” This statement is supported by the results obtained using the carrageenan-induced hind paw edema model in rats. This animal model is used to assess the anti-inflammatory activity resulting from topical drug delivery, that is to say a local response that does not depend on drug concentration into the blood.

Therefore, the authors should clarify the aim of their study (topical or transdermal delivery) modifying the manuscript accordingly.

At line 293: 100 ml should read 100 μl.

minor revision

Author Response

Dear Reviewer,

We would like to express our gratitude for your valuable comments and contributions. We have completed the requested revisions and are resubmitting the manuscript for your reconsideration. Additionally, we have highlighted the changes for easy tracking and provided a comment and response chart to explain the modifications. Thank you for dedicating your precious time to this review.

Hereby, I would like to present the revisions we have made:

A general revison was performed to resolve the conflict of transdermal and topical application.

Comment_1: At line 293: 100 ml should read 100 μl.

Response_1: The authors would like to thank the reviewer for the contribution. The mistake was corrected according to the comment.

Best regards,

Reviewer 3 Report

The manuscript titled “Nanoemulsions as a Promising Carrier for Transdermal Delivery of Etodolac: Formulation Development and Characterization” needs major revisions prior to acceptance

1.      Introduction section needs to be more cohesive. Currently it feels like 3-4 paragraphs just pasted together

2.      Justify in methods- how were CQAs and specifications identified?

3.      Add the retention time in section 2.2.4

4.      Add release kinetics methodology in the methods section

5.      Mention the skin peeling method in section 2.2.5.8.

6.      Mention the skin temperature and TEWL values to determine skin integrity

7.      All figure captions need to be more descriptive. Change all figure captions and add more information

8.      Add a discussion section to discuss all results and to compare the current study with already published studies. Elaborate on how your study improves on the prior studies

Author Response

Dear Reviewer,

We would like to express our gratitude for your valuable comments and contributions. We have completed the requested revisions and are resubmitting the manuscript for your reconsideration. Additionally, we have highlighted the changes for easy tracking and provided a comment and response chart to explain the modifications. Thank you for dedicating your precious time to this review.

Hereby, I would like to present the revisions we have made:

Comment_1: Introduction section needs to be more cohesive. Currently it feels like 3-4 paragraphs just pasted together.

Response_1: The authors would like to thank the reviewer’s comment. We have made the relevant revisions and have highlighted them for your convenience

Comment_2: Justify in methods- how were CQAs and specifications identified?

Response_2: The authors would like to thank the reviewer's comment. CQAs have been carefully defined and relevant citations have been incorporated. The decision was made based on our previous work and using the FMECA analysis we conducted in preliminary studies. Additionally, the FMECA approach has been appropriately cited. In the text, we have cited relevant sources that elucidate the highest risk priority number attributed to the composition of the nanoemulsion.

In the preliminary section (data not shown in the manuscript), we comprehensively discuss the consideration of process parameters (e.g., homogenization rate and duration, sonication amplitude and duration, temperature) and the formulation's composition (including API, surfactant, oil and water ratio, etc.). Our previous data, cited in this section, demonstrates a lower Risk Priority Number (RPN) for process parameters compared to composition. Consequently, these identified CQAs were selected as responses based on the highest RPN value.

Comment_3: Add the retention time in section 2.2.4

Response_3: The authors would like to thank the reviewer for this valuable contribution. The revision was added and highlighted.

Comment_4: Add release kinetics methodology in the methods section

Response_4: The authors would like to thank the reviewer for this valuable contribution. The revision was added and highlighted.

Comment_5: Mention the skin peeling method in section 2.2.5.8.

Response_5: The authors express gratitude to the reviewer for this significant observation. A thorough description of the skin excision procedure was provided by incorporating the methodology from one of our senior colleagues' previous works. The revised section has been emphasized for clarity.

Comment_6: Mention the skin temperature and TEWL values to determine skin integrity

Response_6: The authors would like to explain how skin integrity was decided.

Several methods are available for the determination of skin integrity. Trans epidermal electrical resistance (TEER), trans epidermal water loss (TEWL), and trans epidermal water flux (TWF) were reported as standard methods [1]. The limit values for the TEER, TEWL and TWF were 1 kΩ, 10 g m-2 h-1 and 2.5x 10-3 cm h-1 respectively. Various commercial instruments are presently accessible for measuring Transepidermal Water Loss (TEWL). These instruments find primary applications in in-vivo TEWL measurements, particularly in the domains of dermatological examinations and formulation development. Notably, a recent advancement includes the commercial release of an in-vitro TEWL probe, facilitating initial assessments utilizing excised skin samples [2]. The absence of this type of probe in most research laboratories underscores a constraint of this study. On the other hand, alternative approaches such as visual microscopic examination offer researchers insights into skin integrity. Through a visual examination method adapted from a prior study, it was established that the skin integrity was satisfactory [3].

This explanation was also added to “3.6. In vitro and Ex vivo Release Studies”.

References:

  1. Guth, K.; Schäfer-Korting, M.; Fabian, E.; Landsiedel, R.; van Ravenzwaay, B. Suitability of Skin Integrity Tests for Dermal Absorption Studies in Vitro. Toxicol. Vitr. 2015, 29, 113–123, doi:https://doi.org/10.1016/j.tiv.2014.09.007.
  2. Schoenfelder, H.; Liu, Y.; Lunter, D.J. Systematic Investigation of Factors, Such as the Impact of Emulsifiers, Which Influence the Measurement of Skin Barrier Integrity by in-Vitro Trans-Epidermal Water Loss (TEWL). Int. J. Pharm. 2023, 638, 122930, doi:https://doi.org/10.1016/j.ijpharm.2023.122930.
  3. Klang, V.; Schwarz, J.C.; Haberfeld, S.; Xiao, P.; Wirth, M.; Valenta, C. Skin Integrity Testing and Monitoring of in Vitro Tape Stripping by Capacitance-Based Sensor Imaging. Ski. Res. Technol. 2013, 19, e259–e272, doi:https://doi.org/10.1111/j.1600-0846.2012.00637.x.

Comment_7: All figure captions need to be more descriptive. Change all figure captions and add more information

Response_7: Thanks for the reviewer’s comment. All figure captions checked, and the requests were kindly completed according to the comment of the reviewer.

Comment_8: Add a discussion section to discuss all results and to compare the current study with already published studies. Elaborate on how your study improves on the prior studies

Response_8: Within the manuscript we previously submitted, the results and discussion section contained synthesized information, effectively addressing the core concepts. However, in response to the reviewer's request, we have formulated a dedicated section (Section 3.9: Comparison of current work to previous reports) that comprehensively covers all aspects of the study, offering a more detailed examination. We sincerely appreciate the reviewer's valuable input and contribution to the enhancement of our work.

Best regards,

Round 2

Reviewer 1 Report

I am satisfied with the revised version. Thank you for revising the manuscript positively. 

Reviewer 3 Report

Acceptable